# Early Lyme Borreliosis in Patients Treated with Tumour Necrosis Factor-Alfa Inhibitors

**DOI:** 10.3390/jcm8111857

**Published:** 2019-11-02

**Authors:** Vera Maraspin, Petra Bogovič, Tereza Rojko, Katarina Ogrinc, Eva Ružić-Sabljić, Franc Strle

**Affiliations:** 1Department of Infectious Diseases, University Medical Centre Ljubljana, Japljeva 2, 1525 Ljubljana, Slovenia; vera.maraspin@kclj.si (V.M.); petra.bogovic@kclj.si (P.B.); tereza.rojko@kclj.si (T.R.);; 2Institute of Microbiology and Immunology, Faculty of Medicine, University of Ljubljana, Zaloška 4, 1000 Ljubljana, Slovenia; eva.ruzic-sabljic@mf.uni-lj.si

**Keywords:** Lyme borreliosis, immunocompromised host, TNF-α inhibitors, erythema migrans, treatment, outcome

## Abstract

The study evaluated the course and outcome of erythema migrans in patients receiving tumour necrosis factor-alpha (TNF-α) inhibitors. Among 4157 adults diagnosed with erythema migrans in the period 2009–2018, 16 (2.6%) patients were receiving TNF-α inhibitors (adalimumab, infliximab, etarnecept, golimumab), often in combination with other immunosuppressants, for rheumatic (13 patients) or inflammatory bowel (three patients) disease. Findings in this group were compared with those in 32 sex- and age-matched immunocompetent patients diagnosed with erythema migrans in the same years. In comparison with the control group, the immunocompromised patients had a shorter incubation period (7 vs. 14 days; *p* = 0.0153), smaller diameter of erythema migrans (10.5 vs. 15.5 cm; *p* = 0.0014), and more frequent comorbidities other than immune-mediated diseases (62.5% vs. 25%, *p* = 0.0269), symptoms/signs of disseminated Lyme borreliosis (18.8% vs. 0%, *p* = 0.0324), and treatment failure (25% vs. 0%, *p* = 0.0094). After retreatment with an antibiotic, the clinical course of Lyme borreliosis resolved. Continuing TNF inhibitor treatment during concomitant borrelial infection while using identical approaches for antibiotic treatment as in immunocompetent patients resulted in more frequent failure of erythema migrans treatment in patients receiving TNF inhibitors. However, the majority of treatment failures were mild, and the course and outcome of Lyme borreliosis after retreatment with antibiotics was favourable.

## 1. Introduction

Erythema migrans (EM), the hallmark of early Lyme borreliosis (LB), is a distinct skin manifestation that develops at the site of the *Borrelia*-infected tick bite. The inoculated borreliae may disseminate from the skin to various organs, leading to different manifestations of the disease [1,2,3].

During recent decades, the number of immunocompromised patients has substantially increased and is linked to an augmented risk of infection. A subgroup of individuals with impaired immunity involves those receiving biological therapy with inhibitors of tumour necrosis factor-alpha (TNF-α), a proinflammatory cytokine produced by activated monocytes/macrophages and T-cells. TNF-α has an important role in anti-tumour responses and in acute and chronic inflammation. Antibodies to TNF-α and TNF-α receptor agonists attenuate the inflammatory processes and have been used for treatment of many inflammatory conditions including rheumatoid arthritis, spondylarthritis, psoriatic arthritis, psoriasis, and inflammatory bowel diseases. However, if the immune system protecting from infection is inhibited, the risk of severe infection increases and latent infections may be reactivated [4,5,6].

Only a few series on immunocompromised patients with LB have been published and information on the course and outcome of LB in this group of patients remains incomplete [7,8,9,10]. Data on patients with LB treated with TNF-α inhibitors are limited to individual case reports [11,12,13,14,15,16].

The aim of the present study was to evaluate the course and outcome of EM in adult patients receiving TNF-α inhibitors for their principal disease.

## 2. Patients and Methods

### 2.1. Data Source, Selection of Cases, and Control Subjects

Information was obtained from a database of adult patients diagnosed with EM at the Lyme Borreliosis Outpatient Clinic, Department of Infectious Diseases, University Medical Centre Ljubljana, Slovenia, in the period 2009–2018. The clinical and laboratory data were gathered prospectively using a standardized questionnaire. To be eligible for the study cases needed to fulfil two criteria: (1) presence of EM defined according to European criteria [17] and (2) therapy with TNF-α inhibitors for an underlying disease.

For each patient with EM receiving TNF-α inhibitors, two patients without autoimmune disorders or use of immunosuppressants (immunocompetent patients) diagnosed with EM at our institution in the same year and matched for sex, age and antibiotic treatment were assigned. If in an individual year more than one control EM patient of the same sex and age had been found than the patient with the alphabetically nearest name to the corresponding TNF-α inhibitor recipient was chosen as a control. The only mismatches were in the treatment of three immunocompromised patients who received ceftriaxone whereas their controls were treated with doxycycline. Since we do not treat patients with solitary EM with ceftriaxone unless they have extracutaneous manifestations of LB such as Lyme neuroborreliosis, we were not able to find immunocompetent patients with EM matched for antibiotic treatment.

### 2.2. Clinical Evaluation, Treatment Approach, Definitions

Patients were examined physically and medical histories obtained at enrolment and at follow-up visits two weeks, two months, six months and one year later.

The skin lesion was identified as EM when it: (1) developed days to weeks after a tick bite or after exposure to ticks in a LB-endemic region, (2) appeared as an expanding red or bluish-red plaque, with or without central clearing, and (3) reached >5 cm in diameter. In the case of smaller lesions at presentation, a history of tick bite, a delay in appearance of at least two days, and an expanding erythema at the site of the bite were required for reliable diagnosis. Multiple EM was defined as the presence of two or more skin lesions, at least one of which had to fulfil the size criterion for solitary EM. Particular attention was paid to the appearance of the skin lesion, the presence of associated constitutional symptoms (newly developed or worsened since the onset of the EM and which had no known other medical explanation), and other objective manifestations of LB.

In patients receiving TNF-α inhibitors, EM was treated with doxycycline 100 mg twice daily for 14 days (9 patients), amoxicillin 500 mg three times daily for 15 days (2 patients), azithromycin 500 mg twice daily on the first day followed by 500 mg once daily for four days (2 patients), or ceftriaxone 2 g once daily intravenously for 14 days (3 patients).

Patients were asked to assess the presence of their skin lesion every day and to document when it disappeared. Presence of EM was defined as erythema that could still be seen in daylight and at room temperature.

At the follow-up visit 14 days after the onset of antibiotic treatment patients were asked whether they complied with the treatment, how many tablets (capsules) of antibiotic they still had, and if they had any side effects.

For the present study, treatment failure was defined as: (1) occurrence of objective extracutaneous manifestations of LB, (2) appearance/persistence of subjective symptoms or their increased intensity that could not be attributed to other causes, (3) still visible EM at the follow-up visit 2–3 months after starting antibiotic treatment, or (4) demonstration of borreliae at the site of the previous EM 2–3 months after antibiotic therapy. These patients were re-treated with an alternative antibiotic.

Patients presenting with clinical signs/symptoms of a disseminated form of LB before antibiotic treatment and/or those with treatment failure were interpreted as having a complicated course of EM.

### 2.3. Laboratory and Microbiological Evaluation

Basic laboratory tests (erythrocyte sedimentation rate (ESR), blood cell counts, liver function tests) were performed at the first visit and two weeks later.

Serum immunoglobulin M (IgM) and immunoglobulin G (IgG) antibodies to *Borrelia burgdorferi* sensu lato were measured at baseline and at two-, six-, and 12-month follow-up visits. In the first two years (2009 and 2010), an immunofluorescence assay with a local skin isolate of *Borrelia afzelii* as the antigen was used; titers ≥ 1:256 were considered positive. Later, serum IgM antibodies to outer surface protein C (OspC) and variable-like sequence (VlsE), and IgG antibodies to VlsE borrelial antigens were measured in an indirect chemiluminescence immunoassay (LIAISON, Diasorin, Italy); results were interpreted according to the manufacturer’s instructions [18].

In patients who gave their consent, a punch skin biopsy specimen (3 mm) from the EM border and a whole-blood specimen (9 mL citrated blood) were cultured for borreliae in modified Kelly-Pettenkofer medium. In patients with a positive skin culture result, the biopsy was repeated 2–3 months after the start of antibiotic treatment [18]. Cultures were examined weekly by darkfield microscopy for the presence of borreliae; results were interpreted as negative if no growth was established after 9 weeks for skin and after 12 weeks for blood samples. Identification of borrelial isolates to species level was made using pulsed-field gel electrophoresis after *MluI* restriction of genomic DNA or by PCR-based restriction fragment length polymorphism of the intergenic region [18,19].

### 2.4. Statistical Analyses

Numerical variables were summarized with medians (interquartile ranges, IQR), categorical variables with frequencies and percentages (with 95% confidence intervals). Pretreatment characteristics and the course and outcome of early LB after antibiotic treatment in patients with EM receiving TNF-α inhibitors were compared with the corresponding findings in a control group of previously healthy persons with EM. Categorical variables were compared using the chi-squared test with Yates’ continuity correction or two-tailed Fisher’s exact test; numerical variables were compared using the Mann-Whitney test.

### 2.5. Ethical Considerations

The study was conducted in accordance with the Declaration of Helsinki. The diagnostic and treatment approach used in patients with EM was approved by the Medical Ethics Committee of the Republic of Slovenia (No. 35/05/09 and 145/45/14).

## 3. Results

### 3.1. Basic Pretreatment Clinical Findings in Immunocompromised Patients

During the 10-year period, 16/4157 (2.6%) adult patients diagnosed with typical EM at our institution were receiving TNF-α inhibitors for an underlying disease. Clinical data on the 16 patients are given in Table 1. There were nine women and seven men, with median age 57 (IQR 46.5–61.5) years. Eleven patients were being treated with adalimumab (10 rheumatic disease, 1 Crohn’s disease), three patients with infliximab (two with ulcerative colitis, one with rheumatic disease), one patient with etanercept and a further patient with golimumab (both had rheumatic disease). Six patients were receiving TNF-α inhibitors only, and 10 patients (all with rheumatic disease) had additional treatment with methotrexate (5 patients), leflunomide (3 patients), methylprednisolone (1 patient) or meloxicam (1 patient). Duration of treatment with TNF-α inhibitors prior to development of EM was 9 months to 8 years (median 3 years); all the patients continued with the treatment during the one-year follow-up. Fifteen patients (93.8%) presented with solitary EM, an additional patient (6.3%) with multiple skin lesions (Table 1, patient 14). Two patients with solitary skin lesions reported pronounced newly developed symptoms since the onset of the EM which had no known other medical explanation and were interpreted as being markers of possible borrelial dissemination (Table 1: patients 5 and 13).

### 3.2. Comparison of Immunocompromised and Immunocompetent Patients 

#### 3.2.1. Pretreatment Characteristics 

The basic pretreatment clinical characteristics of the immunocompromised and immunocompetent patients with EM skin lesions are given in Table 2. Comparison of the two groups showed several analogous findings and some distinctions. Similar findings included the frequency of tick bite at the site of later EM, duration of EM prior to diagnosis, increase in EM surface area per day, location and appearance of EM, the frequency of accompanying local and constitutional symptoms, as well as the majority of laboratory results including the isolation rate of borreliae from skin biopsy specimens (Table 1). However, in comparison with immunocompetent patients, those with impaired immunity reported shorter time from a tick bite to the onset of EM (7.5 vs. 14 days; *p* = 0.0153) and had smaller diameter of EM (10.5 vs. 15.5 cm; *p* = 0.0014), but more often had comorbidities other than those for which they were receiving the TNF inhibitor (62.5%, 95% CI: 35.4–84.8 vs. 25%, 95% CI: 11.5–43.4; *p* = 0.0269) and more frequently had symptoms/signs of disseminated LB (18.8%, 95% CI 4.1–45.7 vs. 0%, 95% CI: 0–10.9; *p* = 0.0324), abnormalities at physical examination (37.5%, 95% CI: 15.2–64.6 vs 0%, 95% CI: 0–10.7; *p* = 0.0007), and increased ESR (37.5%, 95% CI: 15.2–64.6 vs. 10.3%, 95% CI: 2.2–27.4; *p* = 0.0499).

No statistically significant difference was found comparing the presence of serum IgM and/or IgG antibodies to borreliae in patients receiving TNF inhibitor and immunocompetent patients (56.3%, 95% CI: 29.9–80.3 vs. 62.5%, 95% CI: 43.7–78.9; *p* = 0.92). However, borrelial IgM antibodies in serum were present more often in patients receiving TNF inhibitors than in the control group (50%, 95% CI: 24.7–75.4 vs. 18.8%, 95% CI: 7.2–36.4; *p* = 0.0421). The isolation rate of borreliae from skin was comparable in the two groups (6/14, 42.9%, 95% CI: 17.7–71.1 vs. 15/29, 51.7%, 95% CI: 32.5–70.6; *p* = 0.83).

#### 3.2.2. Post-treatment Course and Outcome

After the start of antibiotic therapy, the duration of the skin lesions was longer in patients receiving TNF-α inhibitors (median 22, IQR 7–36 days) than in their controls (median 10, IQR 7–20 days), but the difference was not statistically significant. Furthermore, in 3/16 (18.8%, 95% CI: 4.1–45.7%) patients receiving TNF-α inhibitors and in 0/32 (0%, 95% CI: 0–10.9%) controls the duration of erythema exceeded 100 days (*p* = 0.0324).

Treatment failed in 4/16 (25%, 95% CI: 7.3–52.4) patients with impaired immunity, but in none of the control group (0%, 95% CI: 0–10.9; *p* = 0.0094). In three patients with treatment failure, the EM persisted for ≥2 months after starting antibiotic therapy (Table 1: patients 1, 2, 13). In these patients the skin lesions disappeared 35, 40 and 45 days, respectively, after re-treatment with an alternative antibiotic and the subsequent clinical course was smooth. All three patients were seronegative at presentation and remained seronegative during one-year follow-up. The fourth patient with treatment failure (Table 1, patient 4) was a 44-year old man with solitary EM. He had an uneventful course at the 6-month follow-up visit. However, 7 months after beginning antibiotic treatment he developed severe arthralgia, fatigue and back pain. A relapse of rheumatoid arthritis was suspected but was not confirmed by his rheumatologist. At the one-year follow-up, the patient complained of severe symptoms lasting for 5 months and showed an increase of IgG antibodies to VlsE borrelial antigens from 542.1 to 1462.0 AU/mL. He improved clinically within one month after re-treatment with ceftriaxone; the subsequent clinical course during a further one-year follow-up was unremarkable.

A complicated course of LB was found in 6/16 immunocompromised patients (three presented with symptoms/signs of early disseminated LB; four with treatment failure, one of whom had symptoms/signs of early disseminated LB), but in none of the immunocompetent group (37.5%, 95% CI: 15.2–64.6 vs. 0%, 95% CI: 0–10.9; *p* = 0.0007).

## 4. Discussion

There is limited information on the course and outcome of LB in patients with impaired immunity resulting from underlying illness and/or treatment, including therapy with TNF-α inhibitors. These biological drugs are approved for treatment of immune-mediated diseases such as rheumatoid arthritis, psoriatic arthritis, juvenile arthritis, ankylosing spondylitis, psoriasis and inflammatory bowel disease (Crohn’s disease, ulcerative colitis). By reducing inflammation they can ameliorate symptoms, stop disease progression and substantially improve quality of life, enabling greater activity, including activities outdoors with consequently an increased exposure to ticks and development of tick-transmitted diseases [2,3]. Several adverse events have been associated with the use of TNF-α inhibitors, with infections being the most common. The major concern is the increased occurrence of infections, particularly in patients receiving adalimumab or infliximab [20], and the enhanced severity of some bacterial diseases (tuberculosis, pneumonia, listeriosis), viral infections (herpes zoster, hepatitis B and C, cytomegalovirus infection), and invasive fungal infections (histoplasmosis, aspergillosis, cryptococcosis, candidosis) which can be life-threatening [6,21,22,23,24,25]. The American College of Rheumatology therefore recommend that TNF-α inhibitors should not be administered in cases of active bacterial infection or bacterial infection requiring antibiotic therapy [26,27].

In mice, TNF-α appears essential in the immunological control of borrelial infection, and TNF-α blockade may impair elimination of borreliae during antibiotic treatment [28,29], but some of the findings have been challenged [30]. In humans, however, the impact of TNF-α antagonists on the course and outcome of LB is not clear.

A PubMed literature search found no data on the course and outcome of tick-borne diseases such as babesiosis, tick-borne encephalitis or anaplasmosis, and only six reports on individual patients with LB who were receiving TNF-α inhibitors (3 were receiving etanercept, 1 adalimumab, 1 infiximab, 1 certolizumab). Solitary EM was diagnosed in one patient [13], multiple EM in the other [16], Lyme neuroborreliosis in three patients [12,14,15], while one patient presented with lupus-like syndrome and borrelial IgM and IgG antibodies in serum [11]. In five of these six cases the choice of antibiotic was in accord with treatment recommendations for LB (ceftriaxone or doxycycline), while one patient received ceftriaxone and doxycycline concomitantly [16]. Also the dosage and duration of antibiotic therapy was somewhat heterogeneous. The patient with solitary EM [13] was treated with high-dose doxycycline (300 mg/day) for as long as 3 months (according to current recommendations, EM in adults is treated with doxycycline 100 mg twice daily for 14 or even 10 days) and the patients with Lyme neuroborreliosis, patient with multiple EM and the patient with lupus-like syndrome received antibiotics (ceftriaxone 2 patients, doxycycline 2 patients, ceftriaxone and doxycycline 1 patient) for 3 or 2 weeks in standard dosages [11,12,14,15,16]. In all these patients the course and outcome of LB after antibiotic treatment was favourable. In four of the six reported cases, treatment with TNF-α inhibitors was discontinued [12,13,14,16]. In one of these four, the interruption of TNF-α inhibitor treatment (etanercept) resulted in a polyarthritis crisis; the drug was therefore reintroduced [13]. Thus, the reported information was too limited and heterogeneous to reliably answer questions on whether the dosage and length of antibiotic therapy for LB as used for immunocompetent patients is appropriate also for patients receiving TNF-α inhibitors, and whether discontinuation of treatment with TNF-α inhibitors during an ongoing borrelial infection is needed.

In our group of 16 immunocompromised patients with early LB, comparison of pretreatment clinical characteristics, laboratory results and microbiological findings in the immunocompromised patients and the controls revealed analogous findings for the majority but not for all tested parameters (Table 2). Differences in the frequency of abnormalities found at physical examination, increased ESR (which is in Europe very rarely associated with erythema migrans), and probably also more frequent comorbidities other than those for which patients were receiving TNF inhibitors could be attributed to patients’ underlying immune-mediated disease, whereas more frequent demonstration of borrelial serum IgM antibodies was possibly the result of false positivity, as reported in several conditions including inflammatory rheumatism [31,32]. Although we do not have a trustworthy explanation for the shorter incubation period (7.5 vs. 14 days), smaller diameter of EM (10.5 vs. 15.5 cm), and more frequent presence of symptoms/signs indicating or suggesting borrelial dissemination (18.8% vs. 0%) in immunocompromised vs immunocompetent patients, these findings could be related to treatment with TNF-α inhibitors. Since the results are generally in agreement with the immunosuppressant properties of TNF inhibitors, and with their impact on other types of infections [5,6], the differences might offer some insights into the host immune response and the role of TNF. Yet, the interpretation is limited due to the heterogeneity of our group according to underlying illness and immunosuppressive therapy. The finding that all four patients with treatment failure (compared to half of those on TNF inhibitor monotherapy) were receiving methotrexate or leflunomide in addition to TNF-α inhibitor, suggests the impact of immunosuppressive treatment other than TNF inhibition. Nevertheless, it seems that the course of early LB in patients receiving TNF inhibitors differs in some respects from that in immunocompetent patients. However, the long-term outcomes after antibiotic treatment are similar. As reported elsewhere, we have been using the same approach for antibiotic treatment in immunocompromised and immunocompetent patients with LB [7,8,15,33]. In addition, we did not discontinue TNF inhibitors during concomitant borrelial infection. Our initial decision had been to maintain TNF inhibitor treatment in patients with EM (localized LB) but was ambiguous regarding what to do in cases of extracutaneous manifestation of LB such as Lyme neuroborreliosis. In fact, as reported previously, in a patient who developed early Lyme neuroborreliosis (Bannwarth’s syndrome) during treatment of psoriasis with adalimumab the decision was made to temporarily discontinue immunosuppressive therapy. Since the response to antibiotic treatment of LB was favourable, and the underlying illness did not deteriorate, we were pleased with the decision; however, the patient subsequently admitted that she continued to treat herself with adalimumab. Nevertheless, in spite of her continuation of treatment with a TNF inhibitor, the course of Lyme neuroborreliosis was smooth and the outcome one year after treatment was favourable [15]. The present study has shown that using the same antibiotic treatment approach in immunocompromised patients receiving TNF inhibitor as in immunocompetent patients with EM, while continuing the treatment with a TNF-α inhibitor, resulted in more common treatment failure and more often a complicated course of LB in patients receiving TNF-α inhibitor than in the sex- and age-matched immunocompetent patients with EM. However, only one of three patients interpreted as having disseminated LB had objective signs of dissemination, as many as three of four LB treatment failures presented with incomplete disappearance of EM (which is clinically unimpressive failure) while one patient had severe subjective symptoms but without objective clinical findings. Furthermore, all six initially culture positive patients (including one with treatment failure) in whom repeated skin biopsy was performed 2–3 months after antibiotic treatment had a negative borreliae skin culture result, all failures vanished after re-treatment with antibiotics, and the outcome of LB one year after antibiotic re-treatment was favourable.

Patients receiving TNF-α inhibitors had remarkably long persistence of EM after the start of antibiotic therapy (median 22 days in comparison to 10 days in controls; the difference was not statistically significant). Furthermore, in 3/16 (18.8%) immunocompromised patients but in 0/32 immunocompetent patients the duration of erythema was >3 months (*p* = 0.0324). Findings in the control group are in accord with our recent report on EM in immunocompetent adult patients, in which similar approaches were used to assess the course and outcome of EM as in the present study: median time to resolution of EM was 7 days, and the time showed significant prolongation with advancing age; in 11/1176 (0.9%) patients residual erythema could still be seen at the 2–3 month visit [33]. In our previous reports on immunocompromised patients median durations of EM after the beginning of antibiotic treatment were 6 days for patients with solid organ transplantation [8], 7 days for patients having haematological malignancy [10], and 12 days for patients treated with rituximab [34], while the proportions of patients with still visible EM at a visit 2–3 months after institution of antibiotic treatment were 1/6 (17%), 1/53 (1.9%), and 1/7 (14.3%), respectively [8,10,34].

The study has several limitations. In general, due to the approach used in the present study to avoid missing any clinical failures we might have erred on the side of clinical failure: of four LB treatment failures in immunocompromised patients receiving TNF inhibitor three were clinically unimpressive (incomplete disappearance of EM) while the fourth comprised severe subjective symptoms but without objective clinical findings. Because clinicians are typically looking more closely for signs of treatment failure in patients they know to be immunocompromised (and the same is most probably valid for immunocompromised patients themselves) there is a possibility of exaggeration of the observed difference in treatment failures between cases and controls. However, since for more than 30 years in all our patients with EM, regardless of their immune status, the clinical and laboratory data have been gathered prospectively using a standardized questionnaire, chances for such bias are probably negligible. Our immunocompromised patients had heterogeneous underlying illnesses and in several TNF inhibitor was combined with other immunosuppressive drugs, making interpretation of the effects of TNF inhibitor on the course and outcome of early LB more difficult. In addition, although the number of immunocompromised patients receiving TNF inhibitor in the present study was nearly 3-times higher than reported previously [11,12,13,14,15,16] and although identical LB treatment approaches were used for all patients, the number of patients was still too low to enable completely reliable conclusions about the value of these approaches. Nevertheless, our results are probably applicable to European regions with similar ratios of borrelial genospecies causing EM but may not entirely apply to North America, where LB is nearly exclusively caused by *B. burgdorferi* sensu stricto [35].

## 5. Conclusions

Our study has shown that the course of early LB in patients receiving TNF inhibitors, often in combination with other immunosuppressants, somewhat differs from that in immunocompetent patients and that using an identical antibiotic treatment approach as for immunocompetent patients with EM, while continuing treatment with a TNF-α inhibitor, resulted in more common treatment failure and more often a complicated course of LB in patients receiving a TNF-α inhibitor than in sex- and age-matched immunocompetent patients with EM. However, treatment failures were mild and reversible after re-treatment with antibiotics, and the outcome of LB one year after therapy was favourable. Nevertheless, the treatment approach for immunocompromised patients as used in the present study (identical antibiotic treatment approach as for immunocompetent patients with EM while continuing treatment with a TNF-α inhibitor) should be monitored with regular follow-up visits.

## Figures and Tables

**Table 1 jcm-08-01857-t001:** Clinical and epidemiological data on 16 patients who developed solitary erythema migrans during treatment with tumour necrosis factor alpha (TNF-α) inhibitors for their underlying disease.

Patient Number, Sex/AgeYear of EM	Underlying Disease		Erythema Migrans								Isolation of Borreliae from Skin^h^ before Antibiotic/2–3 Months after Antibiotic
	UD/Duration ^a^/AD	Treatment ^b^	Tick-Bite/Incubation ^c^/Duration of EM before Treatment ^d^	Location/Number/Diameters/Appearance	Symptoms Local/Systemic	Antibiotic treatment of EM			Duration after Treatment: Days ^e^ (days ^f^)	Laboratory Results/Serum Antibodies to Borreliae (IgM/IgG) ^g^	
						Initial	Retreatment				
							Reason	Antibiotic			
1 F/57 2009	RA/18 years/AH, HL	Adalimumab 40 mg/2 weeks + methotrexate 15 mg/week	Yes/7/7	Thigh/1/8 × 5 cm/homogeneous	None/none	AZM 1 g day 1, 500 mg days 2–5	Persistence of EM ≥2 months after initial therapy	DOXY 100 mg twice daily for 14 days	105 (35)	Normal/neg/neg	ND/ND
2 F/59 2010	RA/20 years/AH, HL	Adalimumab 40 mg/2 weeks + methotrexate 15 mg/week	No/?/7	Thigh/1/18 × 16 cm/homogeneous	Itching/none	AMX 500 mg three times daily for 15 days	Persistence of EM ≥2 months after initial therapy	DOXY 100 mg twice daily for 14 days	120 (45)	↑ liver enzymes/neg/neg	*Borrelia afzelii*/neg
3 M/55 2013	PA/10 years/AH, HL	Adalimumab 40 mg/2 weeks + methotrexate 12.5 mg/week	No/?/39	Shank/1/21 × 18 cm/homogeneous	Itching, burning/slight headache, arthralgia	DOXY 100 mg twice daily for 14 days	No	-	23	Normal/pos/pos	*B. afzelii*/neg
4 M/44 2013	RA/5 years/None	Adalimumab 40 mg/2 weeks + leflunomide 10 mg/day	No/?/7	Chest/1/11 × 4 cm/ring-like	None/none	DOXY 100 mg twice daily for 14 days	At 7 months: Severe arthralgia, fatigue, back pain	CRO 2 g i.v. once daily for 14 days	3	↑ liver enzymes/pos/pos	neg/ND
5 M/45 2014	RA/6 years/None	Adalimumab 40 mg/2 weeks + leflunomide 10 mg/day	Yes/7/19	Foot/1/12 × 8 cm/homogeneous	None/severe, arthralgia, fatigue, back pain	CRO 2 g iv once daily for 14 days	No	-	14	Normal/pos/pos	*Borrelia garinii/*neg
6 F/60 2013	RA/6 years/None	Adalimumab 40 mg/2 weeks + methotrexate 15 mg/week + methylprednisolone 2 mg/day	Yes/30/3	Abdomen/1/7 × 4 cm/homogeneous	Itching/none	DOXY 100 mg twice daily for 14 days	No	-	21	Normal/pos/pos	*B. afzelii*/neg
7 F/71 2013	PA + PR/10 + 1 years/IDDM	Adalimumab 40 mg/2 weeks + methylprednisolone 4 mg/day	Yes/1/35	Thigh/1/6 × 5 cm/ring-like	None/none	CRO 2 g i.v. once daily for 14 days	No	-	2	↑ ESR/neg/pos	neg/ND
8 F/57 2014	RA/8 years/None	Adalimumab 40 mg/2 weeks + meloxicam 7.5 mg/day	Yes/18/3	Abdomen/1/6 × 4 cm/homogeneous	Burning/none	DOXY 100 mg twice daily for 14 days	No	-	42	↑ ESR/pos/neg	neg/ND
9 M/48 2015	PS/4 years/None	Adalimumab 40 mg/2 weeks	Yes/5/7	Thorax/1/11 × 5 cm/homogeneous	Itching, burning/none	DOXY100 mg twice daily for 14 days	No	-	4	Normal/intermediate/pos	neg/ND
10 F/58 2016	MC/8 years/OP	Adalimumab 40 mg/week	No/?/14	Thigh/1/13 × 8 cm/ring-like	Itching/fatigue	DOXY 100 mg twice daily for 14 days	No	-	30	↑ ESR/pos/neg	*B. afzelii*/neg
11 M/50 2018	PA/10 years/AH	Adalimumab 40 mg/2 weeks	Yes/8/12	Abdomen/1/6 × 3cm/homogenous	Itching/none	DOXY 100 mg twice daily for 14 days	No	-	7	Normal/neg/neg	neg/ND
12 M/33 2011	UC/2 years/None	Infliximab 360 mg/7 weeks	No/?/9	Arm/1/9 × 7 cm/ring-like	Itching/none	AZM 1 g on day 1, 500 mg days 2–5	No	-	28	Normal/neg/neg	neg/ND
13 F/69 2013	RA + PA/25 years/AH, OP, TGD, DS	Infliximab 325 mg/6 weeks + leflunomide 10 mg/day	No/?/21	Arm/1/21 × 18 cm/homogeneous	Burning/fatigue, headache, arthralgia, dizziness	CRO 2 g i.v. once daily for 14 days	Persistence of EM ≥2 months after initial therapy	DOXY 100 mg twice daily for 14 days	110 (40)	↑ ESR, anaemia, ↑ liver enzymes/neg/neg	neg/ND
14 M/36 2016	UC/20 years/None	Infliximab 300 mg/8 weeks	Yes/14/7	Leg/2/12 × 12; 8 × 8 cm/homogenous	None/none	DOXY 100 mg twice daily for 14 days	No	-	7	↑ bilirubin/pos/pos	neg/ND
15 F/63 2015	RA/20 years/TGD, OP	Etanercept 50 mg/week + methotrexate 7.5 mg/week	No/?/9	Abdomen/1/13 × 12 cm/homogenous	Itching/none	DOXY 100 mg twice daily for 14 days	No	-	21	↑ ESR/neg/neg	*B. garinii*/neg
16 F/63 2016	RA+PA/18 years/AH	Golimumab 50 mg/4 weeks	No/?/90	Leg/1/10 × 9 cm/homogenous	None/none	AMX 500 mg three times daily for 15 days	No	-	30	Normal/neg/neg	ND/ND

^a^ Duration of underlying disease prior to diagnosis of EM. ^b^ Treatment of underlying disease at the time of EM. ^c^ Days from tick bite to the onset of erythema migrans (incubation is given for patients who reported a recent tick bite at the site of later EM). ^d^ Days from the onset of erythema migrans (as appreciated by a patient) to diagnosis and initiation of antibiotic treatment. ^e^ Days from the institution of the initial antibiotic treatment to complete resolution of erythema migrans. ^f^ Days from the institution of the second antibiotic treatment to complete resolution of erythema migrans. ^g^ At presentation. ^h^ All patients who had borrelial skin culture also had blood culture; none of the blood cultures were positive for borreliae. TNF = tumour necrosis factor; EM = erythema migrans; UD = underlying disease; AD = additional diseases; F = female; RA = rheumatoid arthritis; AH = arterial hypertension; HL = hyperlipidaemia; AZM = azithromycin; DOXY = doxycycline; neg = negative; ND = not done; ? = unknown; AMX = amoxicillin; ↑ = elevated; *B. = Borrelia*; M = male; PA = psoriatic arthritis; pos = positive; CRO = ceftriaxone; i.v. = intravenously; PR = polymyalgia rheumatica, IDDM = insulin-dependent diabetes mellitus; ESR = erythrocyte sedimentation rate; PS = psoriasis; MC = Morbus Crohn; OP = osteoporosis; UC = ulcerative colitis; TGD = thyroid gland disease; DS = depressive syndrome.

**Table 2 jcm-08-01857-t002:** Comparison of demographic, clinical, laboratory and microbiological data of 16 patients with erythema migrans who were receiving tumour necrosis factor-alpha (TNF-α) inhibitors for their underlying disease, and 32 immunocompetent patients with erythema migrans at the initial visit.

Pretreatment Clinical Characteristics			
	Patients Receiving TNF-Alfa Inhibitor *n* = 16	Immunocompetent Patients *n* = 32	*p*-Value
Age (years)	57 (46.5–61.5)	57 (46.5–61.5)	
Male sex	7 (43.8%)	14 (43.8%)	
Presence of comorbidities	10 (62.5%, 35.4–84.8) *	8 (25%, 11.5–43.4) **	**0.0269**
History of prior LB	5 (31.3%, 11.0–58.7)	5 (15.6%, 5.3–32.8)	0.27
Tick bite ^a^	8 (50%, 24.7–75.4)	13 (40.6%, 23.7–59.4)	0.76
Incubation (days) ^b^	7.5 (5–14)	14 (12–34.5)	**0.0153**
Duration of EM to diagnosis (days)	9 (7–20)	7.5 (5–16)	0.44
Increase in EM surface area per day (cm^2^/day)	4.6 (0.9–7.8)	5.1 (0–12.2)	0.64
Largest diameter of EM (cm)	10.5 (7.5–12.5)	15.5 (12–26)	**0.0014**
Homogenous appearance of EM	12 (75%, 47.6–92.7)	25 (78.1%, 60.0–90.7)	1.00
Location of EM ^c^: extremities trunk	10 (62.5%, 35.4–84.8) 6 (37.5%, 15.2–64.6)	21 (65.6%, 46.8–81.4) 11 (34.4%, 18.6–53.2)	0.92
Local symptoms	10 (62.5%, 35.4–84.8)	18 (56.3%, 37.7–73.6)	0.92
Itching ^d^	8 (50%)	16 (50%)	1.00
Burning ^d^	3 (18.8%)	4 (12.5%)	0.67
Pain ^d^	1 (6.3%)	4 (12.5%)	0.65
Constitutional symptoms Fatigue ^d^ headache ^d^ arthralgia ^d^ myalgia ^d^ dizziness ^d^ fever ^d^	4 (25%, 7.3–52.4) 0 1 (6.3%) 3 (18.8%) 0 1 (6.3%) 0	7 (21.9%, 9.3–40.0) 3 (9.4%) 5 (15.6%) 1 (3.1%) 2 (6.3%) 0 0	1.00 0.0788 0.65 0.10 0.55 0.33
Symptoms/signs of disseminated early LB ^e^	3 (18.8%, 4.1–45.7)	0 (0%, 0–10.9)	**0.0324**
Abnormalities at physical examination	6 (37.5%, 15.2–64.6) ^f^	0 (0%, 0–10.9)	**0.0007**
**Laboratory findings**			
No laboratory abnormalities	2 (12.5%, 1.6–38.4)	14 (43.8%, 26.4–62.3)	0.0657
Increased ESR (>20 mm)	6 (37.5%, 15.2–64.6)	3/29 (10.3%, 2.2–27.4)	**0.0499**
WBC > 10 × 10^9^/L	0	1 (3.1%)	1.00
WBC < 4 × 10^9^/L	0	1 (3.1%)	1.00
Pts < 140 × 10^9^/L	0	0	
Abnormal liver enzymes	9 (56.3%, 29.9–80.3)	14 (43.8%, 26.4–62.3)	0.61
AST	6 (37.5%)	7 (21.9%)	0.31
ALT	6 (37.5%)	10 (31.3%)	0.91
γ-GT	3 (18.6%)	6 (18.8%)	1.00
AP	1 (6.3%)	1 (3.1%)	1.00
**Serology**			
IgM	8 (50%, 24.7–75.4)	6 (18.8%, 7.2–36.4)	**0.0421**
IgG	7 (43.8%, 19.8–70.1)	17 (53.1%, 34.7–70.9)	0.92
IgM and/or IgG	9 (56.3%, 29.9–80.3)	20 (62.5%, 43.7–78.9)	0.76
**Microbiological findings**			
Isolation of borreliae from skin	6 ^g^/14 (42.9%, 17.7–71.1)	15 ^h^/29 (51.7%, 32.5–70.6)	0.83
Isolation of borreliae from blood	0/14 (0%, 0–23.2)	0/29 (0%, 0–11.9)	
**Course and outcome of LB after treatment with antibiotics**			
Duration of EM	22 (7–36)	10 (7–20)	0.0742
Treatment failure	4/16 (25%, 7.3–52.4)	0/32 (0%, 0–10.9)	**0.0094**
Complicated course of LB	6/16 (37.5%, 15.2–64.6)	0/32 (0%, 0–10.9)	**0.0007**

Data are medians (interquartile range) or frequencies (percentage, 95% confidence intervals). *P* values are obtained with the Mann–Whitney test for numerical variables and chi-squared test with Yates’ continuity correction or two-tailed Fisher´s exact test for categorical variables. *p*-Values interpreted as statistically significant (<0.05) are shown in bold. *Data depicted in Table 1. ** Arterial hypertension—6 patients; heart disease—3 patients; diabetes mellitus—2 patients; osteoporosis—1 patient; hyperlipidemia—2 patients. Several patients had more than one comorbidity. ^a^ At the site of later EM skin lesion. ^b^ Data for patients who recalled tick bite at the site of later skin lesion (5 patients on treatment with TNF-α inhibitors and 12 controls did not remember a tick-bite). ^c^ Includes information on the primary lesion for the patient with multiple EM. ^d^ Number (%) of patients with the reported symptom. ^e^ Two patients had severe symptoms associated with EM, 1 had multiple erythema migrans. ^f^ Six patients had clinical findings resulting from underlying illness: 3 patients had slight swelling of small joints of extremities, 3 patients had deformation of small joints of hands and feet. ^g^ Among 6 typed isolates, 4 were *Borrelia afzelii* and 2 *Borrelia garinii*. ^h^ 12/15 isolates were typed, 11 as *B. afzelii*, 1 as *B. garinii*. TNF = tumour necrosis factor; LB = Lyme borreliosis; EM = erythema migrans; ESR = erythrocyte sedimentation rate (normal up to19 mm/h); WBC = white blood cells; Pts = platelets; AST = aspartate aminotransferase (normal serum concentration: <0.58 μkat/L); ALT = alanine aminotransferase (normal serum concentration: <0.74 μkat/L); γ-GT = gamma-glutamyltransferase (normal serum concentration: <0.92 μkat/L); AP = alkaline phosphatase (normal serum concentration: <2.15 μkat/L).

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
