# Peer review of "Early Lyme Borreliosis in Patients Treated with Tumour Necrosis Factor-Alfa Inhibitors"

_jcm, 2019, doi:10.3390/jcm8111857_

Round 1

Reviewer 1 Report

This is an excellent paper on the impact of TNF inhibitors on proven Lyme disease.

While the number of 16 patients with anti-TNF and Lyme disease is inevitably low, the total size of the prospective population of 4157 EM patients is unprecedented, and more extensive data won’t be found anywhere in the future.

The design of a prospective cohort with matched case control is very strong, as are the 1-year follow-up, and the culture confirmation of EM in biopsies in about half of the patients – a complex technique, providing the highest level of evidence.

The paper is methodologically sound, and well-written.

I only have a few minor comments:

Please check text (lines 150-154) versus Table 2. IgM is said to be found more often in the cases but the Table’s p-value is 0.92. I would estimate that 8/16 vs. 6/32 may be significant with a p around 0.025

In contrast, IgG is said to be not different, but the Table’s p-value is 0.04. I assume that 6/16 vs. 17/32 is not statistically different.

In the discussion (lines 250-255), the authors are too modest: “Although we do not have a trustworthy explanation…..”.

It would be fair to delete this, and to state that; “While the number of patients is low, the more rapid course (shorter incubation period), reduced skin inflammation (smaller EM diameter), more frequent dissemination (19% vs 0%) and treatment failures (25% vs 0%) are in agreement with the immunosuppressant properties of anti-TNF, and with its impact on other types of infections, as reported previously”.

The reviewer noted that 4 failure cases were on co-medication with mtx of leflunomide. The authors may want to mention this in the discussion.

In the Methods (lines 58-59), the authors state that 2 patients could not be matched for antibiotic choice in the case control. At first glance, it seems remarkable that no age and sex matched patients could be found who received ceftriaxone among the 4000+ controls. Is this because it is very rare that non-immunocompromized patients would receive ctx as first choice treatment for EM? This would be a reasonable explanation, and it may be helpful to briefly add this to the methods section.

Author Response

This is an excellent paper on the impact of TNF inhibitors on proven Lyme disease. While the number of 16 patients with anti-TNF and Lyme disease is inevitably low, the total size of the prospective population of 4157 EM patients is unprecedented, and more extensive data won’t be found anywhere in the future.

The design of a prospective cohort with matched case control is very strong, as are the 1-year follow-up, and the culture confirmation of EM in biopsies in about half of the patients – a complex technique, providing the highest level of evidence.

The paper is methodologically sound, and well-written.

Response: We would like to thank the reviewer for encouraging words.

I only have a few minor comments:

Please check text (lines 150-154) versus Table 2. IgM is said to be found more often in the cases but the Table’s p-value is 0.92. I would estimate that 8/16 vs. 6/32 may be significant with a p around 0.025

In contrast, IgG is said to be not different, but the Table’s p-value is 0.04. I assume that 6/16 vs. 17/32 is not statistically different.

Response: We are ashamed of this mistake. Corrections done – please see Table 2 of the revised article.

In the discussion (lines 250-255), the authors are too modest: “Although we do not have a trustworthy explanation…..”.

It would be fair to delete this, and to state that; “While the number of patients is low, the more rapid course (shorter incubation period), reduced skin inflammation (smaller EM diameter), more frequent dissemination (19% vs 0%) and treatment failures (25% vs 0%) are in agreement with the immunosuppressant properties of anti-TNF, and with its impact on other types of infections, as reported previously”.

Response: Since one of the other reviewers had suggestion in opposite direction we did not change the original text, however we added a sentence supporting more directly the idea of Reviewer 1. Please see lines 326 –329 of the revised article.

The reviewer noted that 4 failure cases were on co-medication with mtx of leflunomide. The authors may want to mention this in the discussion.

Response: The notion has been included in the discussion. Please see lines 330 –336 of the revised article.

In the Methods (lines 58-59), the authors state that 2 patients could not be matched for antibiotic choice in the case control. At first glance, it seems remarkable that no age and sex matched patients could be found who received ceftriaxone among the 4000+ controls. Is this because it is very rare that non-immunocompromized patients would receive ctx as first choice treatment for EM? This would be a reasonable explanation, and it may be helpful to briefly add this to the methods section.

Response: The reviewer is right - we as a rule do not treat patients with solitary EM with ceftriaxone unless they have extracutaneous manifestation of LB such as Lyme neuroborreliosis. This is why we were not able to find immunocompetent patients with EM matching for antibiotic treatment.

Reviewerˊs suggestion has been appreciated. Please see lines 81–83 of the revised article.

Reviewer 2 Report

This is a well done study on an important topic, namely how do Lyme patient fair if they are also on TNF inhibitors.  The authors are clear about their study design and the limitations of the work.   Comments are below:

1.  Lines 152-154 describe the data for IgM antibodies for the two group but the text refers to a p value of p=0.0421 - but if you look at Table 2, this p value is associated with the IgG antibody data - is this an error?

2.  line 80 , do the authors mean "complied" vs "complained"

3.  The authors might speculate a bit beyond " it differs in some respects" (line 255) as to why there was an increase in treatment failure among the TNF group that required re-treatment?   Does this offer some insight into the host immune response and the role of TNF?

Author Response

Comments and Suggestions for Authors

This is a well done study on an important topic, namely how do Lyme patient fair if they are also on TNF inhibitors.  The authors are clear about their study design and the limitations of the work.   

Comments are below:

 Lines 152-154 describe the data for IgM antibodies for the two group but the text refers to a p value of p=0.0421 - but if you look at Table 2, this p value is associated with the IgG antibody data - is this an error?

Response: We are ashamed of this mistake. Corrections done – please see Table 2 of the revised article.

 line 80 , do the authors mean "complied" vs "complained"

Response: Corrected – please see line 106 of the revised article.

 The authors might speculate a bit beyond " it differs in some respects" (line 255) as to why there was an increase in treatment failure among the TNF group that required re-treatment?   Does this offer some insight into the host immune response and the role of TNF?

Response: We added a speculation according to the reviewerˊs suggestion – please see lines 326–329 of the revised article.

Reviewer 3 Report

None, excellent work.

Author Response

Response: Many thanks!

Reviewer 4 Report

This retrospective case-control study reports potential differences in the clinical presentation and treatment response of EM between patients treated with TNF inhibitors and matched unexposed controls. While the number of cases is small, you report several prominent differences between groups, although the ultimate clinical outcomes are reassuringly similar between groups. The findings are interesting, but the clinical details reported and absence of a TNF inhibitor-unexposed group with similar underlying diseases raise questions about the clinical significance of the findings.

Major:
1. Confounding by indication: One main question is whether the group differences relate to treatment with TNF inhibitors or the influence of having an underlying autoimmune disease. This question is not easily answered with the current design. If feasible, I would consider including another comparator group of patients (preferably similar in matching characteristics) who had underlying autoimmune diseases for which TNF inhibitors were indicated (e.g., RA, PA, IBD, or psoriasis) but not used. Comparisons between the cases and this alternative group would help distinguish whether some of the differences attributed to TNF inhibitors were in fact due to the presence of underlying autoimmune disease. Also, it would be helpful to note at least qualitatively whether the underlying autoimmune disease was active at the time of presentation with EM, which could explain things like elevated ESR.

2. Choice of controls: In any case-control study, the approach to choosing controls strongly influences the validity of the comparisons between groups. Readers would benefit from seeing more details about controls were selected - presumably from a database of patients seen in this clinic, but based on what criteria? For example, in selecting "immunocompetent" individuals, please indicate whether or not controls were screened for features other than exposure to TNF inhibitors, such as presence of autoimmune disease, baseline treatment with other medications (e.g., non-TNFi immunosuppressants), or features of the clinical EM presentation (e.g., solitary vs. multiple). Since other conditions could lead to an "immunocompromised" state (e.g., diabetes, RA without TNF inhibition), and we do know whether these conditions were present in controls, the use of "TNF inhibitor-exposed" or "biologic-exposed" might be a more appropriate descriptor of the case group. Finally, for greater transparency, please specify the comorbidities present in each group in Table 2.

3. Interpretation of clinical events: As described in the discussion (lines 273-276), several clinical differences between groups at baseline and in follow-up seemed to be somewhat subjective: for example, whether the "severe" baseline symptoms in cases 5 and 13 were due to disseminated Lyme disease or the underlying rheumatic disease; and whether the clinical events observed for case 4 might related not to treatment failure but a repeat primary Lyme infection. I might consider a sensitivity analyses repeating the comparisons if these subjective findings were interpreted differently (i.e., not due to disseminated Lyme disease and treatment failure, respectively). I would also move lines 273-276 to the limitations section.

Minor:
Abstract: "the clinical course of Lyme borreliosis was smooth" might be stated more directly, eg, "symptoms of LB resolved" or the like.

Methods:
Line 53: change "patients" to "cases"

Line 112: Yates' continuity correction is fine but sometimes viewed as too conservative for small cells sizes. Consider instead using Fisher's exact test for comparisons with expected cell sizes <5.

Results:
Table 1: Please unbold the first row of data. Also spell out or define Mb for "Mb Crohn" in legend.

Discussion:
Lines 206-9: "By reducing inflammation...tick-transmitted diseases": The increased incidence of LB by the mechanism of increased activity and exposure is an interesting and reasonable hypothesis that is not directly tested in this study. Please cite evidence for this statement or otherwise soften this statement (suggest that this could occur) and focus on the increased risk of infections with TNF inhibition.

Author Response

Comments and Suggestions for Authors

This retrospective case-control study reports potential differences in the clinical presentation and treatment response of EM between patients treated with TNF inhibitors and matched unexposed controls. While the number of cases is small, you report several prominent differences between groups, although the ultimate clinical outcomes are reassuringly similar between groups. The findings are interesting, but the clinical details reported and absence of a TNF inhibitor-unexposed group with similar underlying diseases raise questions about the clinical significance of the findings.

Major

Confounding by indication: One main question is whether the group differences relate to treatment with TNF inhibitors or the influence of having an underlying autoimmune disease. This question is not easily answered with the current design. If feasible, I would consider including another comparator group of patients (preferably similar in matching characteristics) who had underlying autoimmune diseases for which TNF inhibitors were indicated (e.g., RA, PA, IBD, or psoriasis) but not used. Comparisons between the cases and this alternative group would help distinguish whether some of the differences attributed to TNF inhibitors were in fact due to the presence of underlying autoimmune disease.

Response: Among limitations of the present study we exposed that the number of patients was small, that patients had heterogeneous underlying illnesses and that in several patients TNF inhibitor was combined with other immunosuppressive drugs, making interpretation of the effects of TNF inhibitor on the course and outcome of early LB more difficult. However, although we were not able to reliable distinguish between the effect of TNF inhibitors, other immunosuppressive drugs and heterogeneity of underlying illnesses, our findings are valid for the group of patients characterized with the presence of underlying autoimmune disease treated with TNF inhibitors of whom the majority were receiving also other immunosuppresive drugs.

Unfortunately we are not in the position to include another control group, however, we further emphasized the drawback exposed by the reviewer in the text of the revised article (please see discussion, lines 329–336).

Also, it would be helpful to note at least qualitatively whether the underlying autoimmune disease was active at the time of presentation with EM, which could explain things like elevated ESR.

Response: Since erythema migrans is (at least in Europe) only very rarely associated with mild elevation of routine markers of inflammation (such as ESR and CRP), the finding of elevated ESR suggests other reasons. Thus, patients shown in Table 1 with elevated ESR most probably had “active” underlying disease. This interpretation has been added to the revised version of the article – please see discussion, lines 323–324 of the revised article. 

Choice of controls: In any case-control study, the approach to choosing controls strongly influences the validity of the comparisons between groups. Readers would benefit from seeing more details about controls were selected - presumably from a database of patients seen in this clinic, but based on what criteria? For example, in selecting "immunocompetent" individuals, please indicate whether or not controls were screened for features other than exposure to TNF inhibitors, such as presence of autoimmune disease, baseline treatment with other medications (e.g., non-TNFi immunosuppressants), or features of the clinical EM presentation (e.g., solitary vs. multiple). Since other conditions could lead to an "immunocompromised" state (e.g., diabetes, RA without TNF inhibition), and we do know whether these conditions were present in controls, the use of "TNF inhibitor-exposed" or "biologic-exposed" might be a more appropriate descriptor of the case group.

Response: We defined the approaches used for the selection of our control group in more detail. Please see Methods, lines 77–79.

 Finally, for greater transparency, please specify the comorbidities present in each group in Table 2.

Response: Underlying illnesses in addition to immunocompromised condition had been given in the submitted article for the group receiving TNF inhibitor (Table 1) and  have been shown for the control group in the revised article – please see Table 2 of the revised article. 

Interpretation of clinical events: As described in the discussion (lines 273-276), several clinical differences between groups at baseline and in follow-up seemed to be somewhat subjective: for example, whether the "severe" baseline symptoms in cases 5 and 13 were due to disseminated Lyme disease or the underlying rheumatic disease; and whether the clinical events observed for case 4 might related not to treatment failure but a repeat primary Lyme infection. I might consider a sensitivity analyses repeating the comparisons if these subjective findings were interpreted differently (i.e., not due to disseminated Lyme disease and treatment failure, respectively).

Response: Cases 5 and 13 had “severe” symptoms which appeared with the onset of skin lesion and disappeared during antibiotic treatment; in addition in one of the cases levels of the routine inflammatory markers were in normal range (Table 1).

We are rather certain that “repeat primary Lyme infection” manifested solely with symptoms - without (typical) clinical signs - is a rather elusive diagnosis which is better to avoid.  

Thus, since we feel strongly that chances for alternative interpretation as proposed by the reviewer are much smaller than the original explanation, we would prefer not to do comparisons according to changed interpretation.

I would also move lines 273-276 to the limitations section.

Response: The statement that “… due to the approach used in the present study to avoid missing any clinical failures we might have erred on the side of clinical failure” has been extended. Please see paragraph on limitations, lines 386-389.  

Minor:
Abstract: "the clinical course of Lyme borreliosis was smooth" might be stated more directly, eg, "symptoms of LB resolved" or the like.

Response: Performed. Please see line 34 of the revised article.

Methods:
Line 53: change "patients" to "cases"

Response: Performed. Please see line 75 of the revised article.

Line 112: Yates' continuity correction is fine but sometimes viewed as too conservative for small cells sizes. Consider instead using Fisher's exact test for comparisons with expected cell sizes <5.

Response: In fact, for expected cell sizes <5 Fisher's exact test was performed. The information has been added. Please see Methods, lines 151–152 of the revised article.

Results:
Table 1: Please unbold the first row of data. Also spell out or define Mb for "Mb Crohn" in legend.

Response: Performed. Please see Table 1 of the revised article.

Discussion:
k Lines 206-9: "By reducing inflammation...tick-transmitted diseases": The increased incidence of LB by the mechanism of increased activity and exposure is an interesting and reasonable hypothesis that is not directly tested in this study.

Please cite evidence for this statement or otherwise soften this statement (suggest that this could occur) and focus on the increased risk of infections with TNF inhibition.

Response: While it has been well recognized that exposure to ticks is an essential prerequisite to acquire tick-transmitted disease (according to reviewer´s suggestion we added references for this statement – please see lines 281–282), there is no (direct and reliable) data that immunocompromised patients have increased risk to acquire a tick-transmitted infection infection and illness in comparison to immunocompetent persons. In the present study we did not assess the risk of infection but the course and outcome of tick-transmitted disease in immunocompromised patients receiving TNF inhibitors.

Round 2

Reviewer 4 Report

The changed made in this version have strengthened the paper and its message. I have a few additional suggestions for clarity:

Line numbers refer to the version jcm-613228clean

Methods:

Confounding by indication: Your decision not to add another control group is reasonable, but I would add to the abstract a phrase alluding to this confounding, e.g., line 24-25: "patients _with autoimmune diseases_ receiving TNF inhibitors, _often in combination with other immunosuppressants_." (The phrase "than in immunocompetent patients" can probably be removed given the earlier phrase "as in immunocompetent patients.") Similarly, in line 347 of conclusion, consider "patients _with autoimmune diseases_ receiving TNF inhibitors, _often in combination with other immunosuppressants_" For clarity, in the methods I would explicitly describe the disease and drug parameters that you used to select immunocompetent controls, such as patients without autoimmune disorders or use of immunosuppressants. Your additional text (lines 284-290) nicely sum the potential contribution of immunosuppressants. The statistical comparison between groups means little with such small numbers. I would suggest removing lines 290-293 and adding a phrase to the prior sentence such as "compared to half of those on TNFi monotherapy."

Terminology of participants: it appears that the word "patients" is sometimes used synonymously with "cases" (e.g., line 49, "Data Source, Selection of Patients and Control Subjects", line 53, "To be eligible for the study patients needed to fulfill two criteria", line 324-5, "Our patients had heterogeneous underlying illnesses"). I find this confusing, as the controls were also technically patients in the clinic. Adding to the confusion is the term "cases" to refer to controls (line 55, "two immunocompetent cases diagnosed with...") For clarity, I would suggest being more consistent in the terminology used: "cases" or "immunocompromised patients" or the like (not just "patients" without descriptor) vs. "controls" or "immunocompetent patients" or the like.

Interpretation of clinical events: your response regarding treatment failures is well taken and reasonable. Nonetheless, I am still left wondering whether clinicians themselves are (understandably) looking more closely for signs of treatment failure in patients they know to be immunocompromised. The same could be said for immunocompromised patients themselves. Thus, I would elaborate: "we might have erred on the side of clinical failure _among cases_:..." After the inserted text ("...but without objective clinical findings") I would also add a line acknowledging the possibility of ascertainment bias that may have exaggerated the observed difference in treatment failures between cases and controls.

Author Response

The changed made in this version have strengthened the paper and its message. I have a few additional suggestions for clarity:

Line numbers refer to the version jcm-613228clean

Methods:

Confounding by indication: Your decision not to add another control group is reasonable, but I would add to the abstract a phrase alluding to this confounding, e.g., line 24-25: "patients _with autoimmune diseases_ receiving TNF inhibitors, _often in combination with other immunosuppressants_." (The phrase "than in immunocompetent patients" can probably be removed given the earlier phrase "as in immunocompetent patients.")

Response: Performed. Please see lines 26 and 37 of the revised version of the article (jcm-613228clean).

Similarly, in line 347 of conclusion, consider "patients _with autoimmune diseases_ receiving TNF inhibitors, _often in combination with other immunosuppressants_"

Response: Performed. Please see line 394.

For clarity, in the methods I would explicitly describe the disease and drug parameters that you used to select immunocompetent controls, such as patients without autoimmune disorders or use of immunosuppressants. Your additional text (lines 284-290) nicely sum the potential contribution of immunosuppressants.

Response: Performed. Please see lines 74–75.

The statistical comparison between groups means little with such small numbers. I would suggest removing lines 290-293 and adding a phrase to the prior sentence such as "compared to half of those on TNFi monotherapy."

Response: Performed. Please see lines 325–328.

Terminology of participants: it appears that the word "patients" is sometimes used synonymously with "cases" (e.g., line 49, "Data Source, Selection of Patients and Control Subjects", line 53, "To be eligible for the study patients needed to fulfill two criteria", line 324-5, "Our patients had heterogeneous underlying illnesses"). I find this confusing, as the controls were also technically patients in the clinic. Adding to the confusion is the term "cases" to refer to controls (line 55, "two immunocompetent cases diagnosed with...") For clarity, I would suggest being more consistent in the terminology used: "cases" or "immunocompromised patients" or the like (not just "patients" without descriptor) vs. "controls" or "immunocompetent patients" or the like.

Response: Performed. Please see lines 67, 71, 74–75, 186, 308, 394.

Interpretation of clinical events: your response regarding treatment failures is well taken and reasonable. Nonetheless, I am still left wondering whether clinicians themselves are (understandably) looking more closely for signs of treatment failure in patients they know to be immunocompromised. The same could be said for immunocompromised patients themselves. Thus, I would elaborate: "we might have erred on the side of clinical failure _among cases_:..." After the inserted text ("...but without objective clinical findings") I would also add a line acknowledging the possibility of ascertainment bias that may have exaggerated the observed difference in treatment failures between cases and controls.

Response: As suggested by the reviewer we included this potential limitation in the text of the revised version of our article. Please see lines 374–381.

We would like to thank the reviewer for stimulating and guiding us to further improve the report.